# Histone deposition pathways determine the chromatin landscapes of H3.1 and H3.3 K27M oncohistones

Jay F Sarthy[1,2], Michael P Meers[1], Derek H Janssens[1], Jorja G Henikoff[1], Heather Feldman[3], Patrick J Paddison[3], Christina M Lockwood[4], Nicholas A Vitanza[2,5], James M Olson[2,5], Kami Ahmad[1]*, Steven Henikoff[1,6]*

[1]Basic Sciences Division, Fred Hutchinson Cancer Research Center, Seattle, United States; [2]Cancer and Blood Disorders, Seattle, United States; [3]Human Biology Division, Fred Hutchinson Cancer Research Center, Seattle, United States; [4]Department of Laboratory Medicine and Pathology, University of Washington School of Medicine, Seattle, United States; [5]Clinical Research Division Fred Hutchinson Cancer Research Center, Seattle, United States; [6]Howard Hughes Medical Institute, Chevy Chase, United States

*For correspondence:
kahmad@fredhutch.org (KA);
steveh@fhcrc.org (SH)

Competing interests: The authors declare that no competing interests exist.

**Abstract** Lysine 27-to-methionine (K27M) mutations in the H3.1 or H3.3 histone genes are characteristic of pediatric diffuse midline gliomas (DMGs). These oncohistone mutations dominantly inhibit histone H3K27 trimethylation and silencing, but it is unknown how oncohistone type affects gliomagenesis. We show that the genomic distributions of H3.1 and H3.3 oncohistones in human patient-derived DMG cells are consistent with the DNAreplication-coupled deposition of histone H3.1 and the predominant replication-independent deposition of histone H3.3. Although H3K27 trimethylation is reduced for both oncohistone types, H3.3K27M-bearing cells retain some domains, and only H3.1K27M-bearing cells lack H3K27 trimethylation. Neither oncohistone interferes with PRC2 binding. Using *Drosophila* as a model, we demonstrate that inhibition of H3K27 trimethylation occurs only when H3K27M oncohistones are deposited into chromatin and only when expressed in cycling cells. We propose that oncohistones inhibit the H3K27 methyltransferase as chromatin patterns are being duplicated in proliferating cells, predisposing them to tumorigenesis.

## Introduction

Diffuse midline gliomas (DMGs) are lethal pediatric brain tumors associated with mutations in genes encoding either histone H3.1 or H3.3, most frequently the lysine 27-to-methionine (H3K27M) 'onco-histone' substitutions (*Schwartzentruber et al., 2012*; *Wu et al., 2012*). H3K27M oncohistones cause the global reduction of histone H3 lysine 27 trimethylation (H3K27me3), a covalent modification that marks silenced regions in the genome. Since H3K27M oncohistones comprise only ~5–15% of the total H3 histone within DMG cells (*Chan et al., 2013*; *Lewis et al., 2013*), they are thought to dominantly inhibit the H3K27 methyltransferase Enhancer of Zeste Homologue-2 (EZH2). Indeed, an H3K27M peptide binds in the active site of EZH2 and inhibits catalytic activity in vitro, and ectopic expression of either oncohistone inhibits H3K27 methylation in vivo (*Chan et al., 2013*; *Lewis et al., 2013*; *Bender et al., 2013*; *Justin et al., 2016*; *Stafford et al., 2018*). These studies have led to the hypothesis that oncohistones inhibit Polycomb-mediated repression of oncogenes, predisposing cells to tumorigenesis.

Most DMG patients carry K27M mutations in the genes encoding the histone variant H3.3, and only ~20% of mutations are in histone H3.1 genes (*Wu et al., 2012*; *Fontebasso et al., 2014*). This

ratio is surprising given that there are only two genes encoding H3.3 and 12 genes encoding H3.1 in the human genome. In addition, although these two histones are very similar, H3.3 or H3.1 K27M mutations are associated with distinct sets of secondary mutations in cancers, and H3.1 mutations are associated with earlier onset gliomagenesis. Finally, while the H3.3 mutations are restricted to gliomas, H3.1 mutations have also been identified in AML and melanomas (*Mackay et al., 2017*; *Nacev et al., 2019*; *Lehnertz et al., 2017*). These differences suggest that the two oncohistones differ in tumorigenic effects.

Studies in mammalian cells suggest that both H3.1K27M and H3.3K27M oncohistones require cell cycle progression in order to inhibit EZH2 (*Chan et al., 2013*; *Nagaraja et al., 2019*), although the etiology of this dependency remains unknown. Histone H3.1 is massively produced only in S phase of the cell cycle, but histone H3.3 is produced constitutively. While the bulk of histone deposition occurs during DNA replication as new chromatin is assembled, the H3.3 histone is deposited both during DNA replication and at sites of active histone turnover, and these are evolutionarily conserved properties of the two histone types (*Ahmad and Henikoff, 2002*; *Tagami et al., 2004*; *Drané et al., 2010*; *Ray-Gallet et al., 2011*; *Clément et al., 2018*). Capitalizing on this conservation, we use *Drosophila* to show that overexpressing either H3K27M oncohistone inhibits H3K27 methylation only in cells progressing through S-phase and only if deposited into chromatin. To directly assess the genomic distribution of H3.3 and H3.1 K27M oncohistones, we applied CUT&RUN chromatin profiling (*Skene and Henikoff, 2017*) to a panel of patient-derived DMG cell lines. We demonstrate that the H3.1 K27M oncohistone is distributed across the genome, consistent with replication-coupled deposition, and these cells have very low H3K27 methylation throughout the genome. In contrast, the bulk of H3.3 K27M oncohistone localizes to sites of active histone turnover, although we also detect the oncohistone at a low level genome-wide, which is consistent with H3.3 deposition during DNA replication. While H3.3K27M-bearing cells have low global levels of H3K27 methylation, they retain high level methylation at a small number of domains. Finally, we find that neither H3K27M oncohistone interferes with PRC2 binding to chromatin in DMG cells. These results support a model where H3K27M oncohistones inhibit PRC2 on chromosomes, helping to explain the origin of gliomas during proliferative periods in development and the spectra of secondary mutations in these gliomas.

## Results

### Chromatin-bound K27M histone inhibits H3K27 trimethylation in cycling cells

Histone H3 variants are highly conserved across evolution, and identical H3.3 histones are produced in both humans and *Drosophila* (*Ahmad and Henikoff, 2002*). Humans have two replication-dependent H3-type histones – H3.1 and H3.2 – while *Drosophila* has only one, which is identical to H3.2. Therefore, to dissect the inhibition of H3K27 methylation by oncohistone variant types, we used *Drosophila* cell and animal models. We first transfected *Drosophila* S2 cells to overexpress FLAG epitope-tagged wild-type or H3K27M oncohistone constructs, and allowed cells to progress through two to three cell cycles with expression of the transfected constructs. Nuclei that overexpress tagged histone H3.2 or H3.3 show broad staining for H3K27 trimethylation at similar levels as untransfected control nuclei (*Figure 1A,B*). In contrast, the same constructs with a K27M mutation show dramatic reduction of H3K27me3 (*Figure 1A,B*). These results show that both H3.2 and H3.3 K27M oncohistones can inhibit H3K27 methylation to similar degrees, at least when similarly overexpressed.

A soluble H3K27M tail peptide can inhibit the EZH2 methyltransferase in vitro (*Lewis et al., 2013*), so we introduced a construct encoding only the histone H3.2 N-terminal tail (residues 1–44) fused to Red Fluorescent Protein (RFP). This H3 tail protein cannot incorporate into nucleosomes as it lacks a histone fold domain, but a large fraction of it localizes within the nucleus (*Figure 1A*). We observed that H3K27 trimethylation levels are unaffected in cells expressing either the wild-type histone tail or the K27M histone tail fusion protein (*Figure 1A,B*). Thus, we infer that while the H3K27M oncohistone can block H3K27 trimethylation, it must be incorporated into chromatin to do so.

Incorporation of H3.1 and H3.2 histones into chromatin is coupled to DNA replication as new nucleosomes are assembled behind replication forks (*Ahmad and Henikoff, 2002*; *Tagami et al.,*

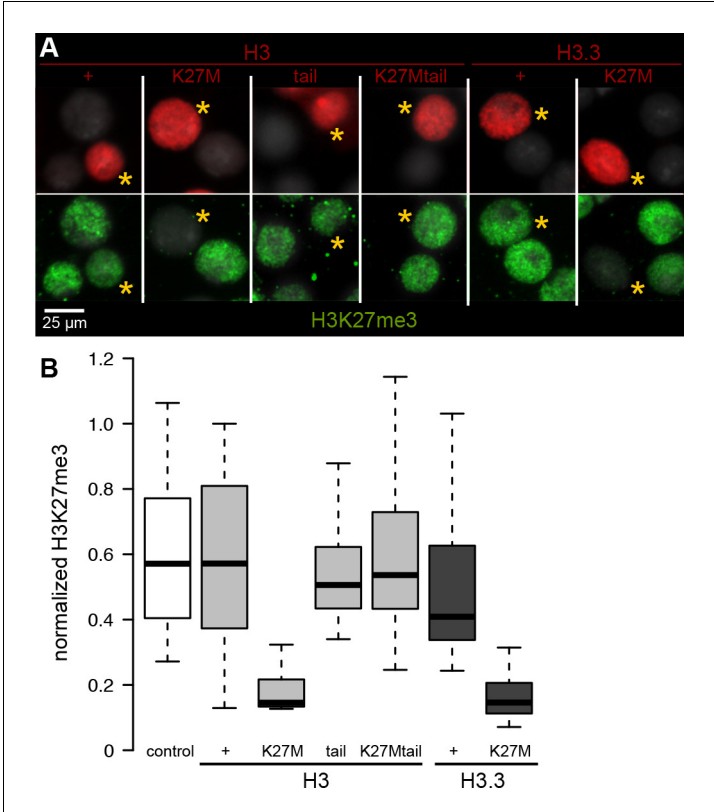

**Figure 1.** Chromatin-bound K27M histones inhibit H3K27 trimethylation. *Drosophila* S2 cells were transfected with epitope-tagged histone constructs, and immunostained for H3K27 trimethylation (green) after 2 days of protein (red) expression. (A) Representative images of non-transfected cells and cells transfected with the indicated epitope-tagged histone construct (yellow asterisks). (B) The mean signal intensity of 50 transfected nuclei and of 50 non-transfected nuclei from two transfections for each construct is plotted.

*2004*). Expression of a H3.3K27M mutant histone in developing wing imaginal discs inhibits H3K27 trimethylation and Polycomb-mediated silencing (*Herz et al., 2014*; *Ahmad and Spens, 2019*). To determine if the H3K27M oncohistone can inhibit methylation in both cycling and in non-dividing cells, we used the developing *Drosophila* eye as a system where we could control induction of histone proteins during the last developmentally-instructed cell division. Following proliferation of eye progenitor cells, one last synchronized wave of cell division moves across a disc just posterior to the morphogenetic furrow (MF) (*Figure 2A*). The lateeye-specific *GMR-GAL4* driver produces the GAL4 activator only in the posterior portion of the disc, and thus induction of a GAL4-responsive histone transgene will produce the protein only in cells that are destined to go through only one more division before terminally differentiating into photoreceptors (*Figure 2A*). Since we found that H3 and H3.3 oncohistones behave similarly when overexpressed in *Drosophila* cell culture, we used this GAL4 system to overexpress wild-type H3.3 or H3.3K27M histones from identical transgene constructs inserted at the same landing site in the genome, so that we could directly compare their effects on H3K27 trimethylation. Expression of either H3.3 or H3.3K27M at this late time in development had no effect on eye morphology (*Figure 2C,D*). Eye imaginal discs from wild-type larvae show similar staining of H3K27 trimethylation in both the anterior and posterior differentiating regions (*Figure 2I*), and induction of a wild-type H3.3 transgene had no effect on this pattern (*Figure 2J*). In contrast, induction of the H3.3K27M oncohistone dramatically inhibits H3K27 trimethylation posterior to the morphogenetic furrow in the eye disc (*Figure 2K*).

Studies in mammalian cells have suggested that cells must progress through at least one cell cycle before K27M oncohistones can inhibit the EZH2 methyltransferase (*Chan et al., 2013*; *Nagaraja et al., 2019*). To determine if inhibition in *Drosophila* also requires cell proliferation, we induced the H3.3K27M transgene and co-induced the cyclin E inhibitor p21, which blocks the last S

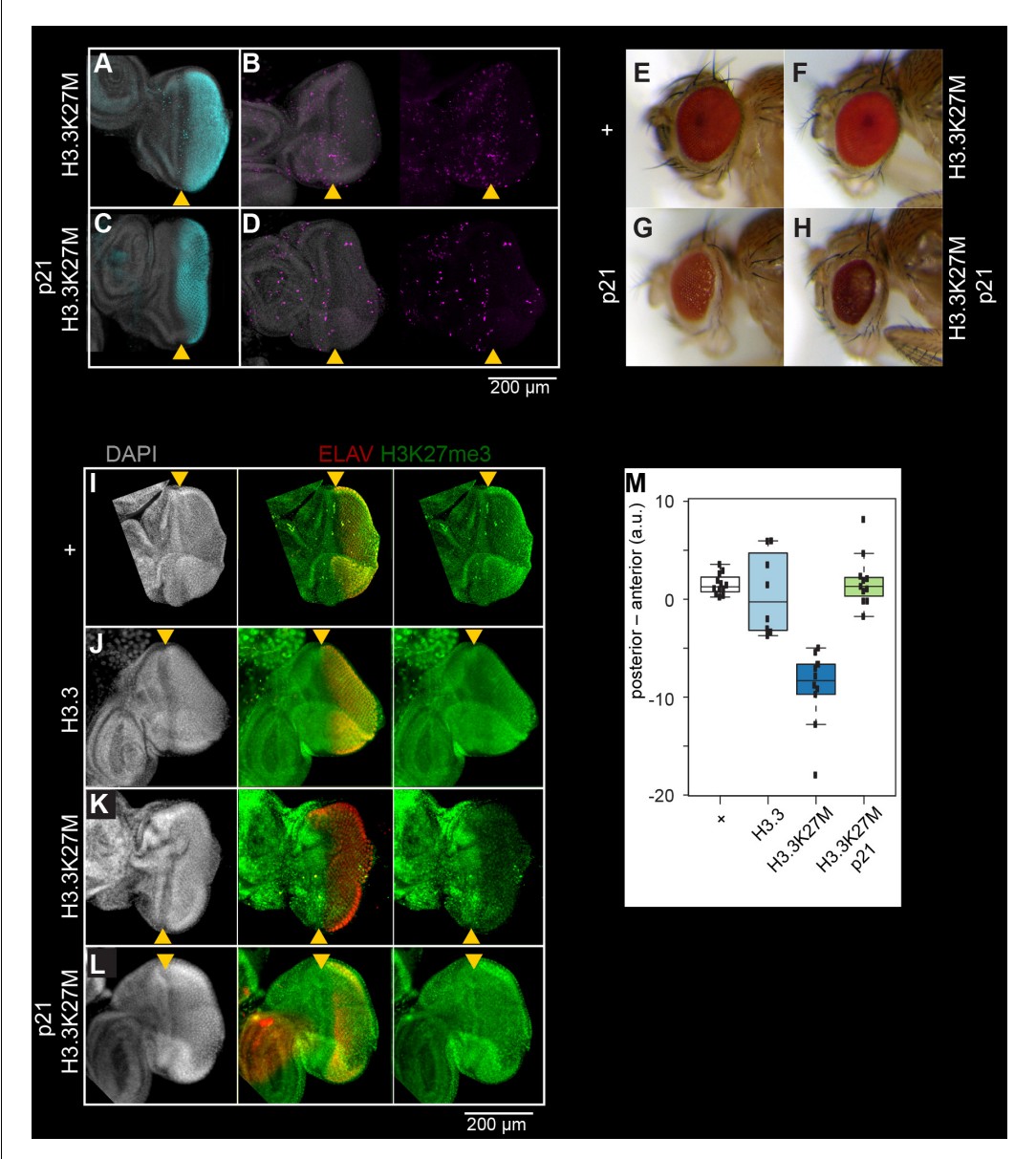

**Figure 2.** Inhibition of H3K27 methylation by K27M histones is limited to cycling cells. Eye imaginal discs from late larvae are divided by the morphogenetic furrow (MF, yellow arrowhead) into an anterior region (left) with asynchronously dividing cells and a posterior region (right). Discs were immunostained for the ELAV neuronal marker (red), which marks differentiating photoreceptors in the posterior region of the disc, the H3K27me3 modification (green), the H3.3K27M oncohistone (blue), or for mitotic cells (S10-phosphorylation, magenta). (A,B) In eye imaginal discs, one last wave of mitosis occurs in the posterior portion (to the right) of the disc behind the MF, as cells initiate neuronal differentiation. The *GMR-GAL4* driver induces expression of the H3.3K27M oncohistone in this region of the disc. (C,D) Expression of the S phase inhibitor p21 in the posterior region of the eye disc blocks progression of the last cell cycle, indicated by the absence of mitotic cells in the posterior region of the eye disc. *GMR-GAL4* induced expression of the H3.3K27M oncohistone in the posterior region of the disc is unaffected. (E,F) Adult eye morphology is unaffected by expression of either a wild-type H3.3 transgene (+) or an H3.3K27M transgene late in development by the *GMR-GAL4* driver. (G,H) GMR-induced expression of the cell cycle inhibitor p21 results in moderately reduced eye size in both wild-type (G) and H3.3K27M-expressing animals (H). (I) Eye imaginal discs from wild-type larvae (+) show high H3K27me3 signal apparent in both the anterior and posterior regions of the eye disc. (J) Induced overexpression of a wild-type H3.3 histone in the posterior portion of the eye disc does not affect H3K27me3 staining. (K) Induced overexpression of H3.3K27M histone strongly reduces H3K27me3 staining in the posterior region of the eye disc. (L) Co-expression of H3.3K27M and the p21 inhibitor show high level H3K27me3 staining in both the anterior and posterior regions of the eye disc. (M) Quantification of H3K27me3 signal intensity differences between the anterior portion of eye discs and the posterior portion, where the *GMR-GAL4* driver induces histone transgene expression. At least 10 discs were measured for each genotype.

phase in the eye disc and arrests cell division (*Ollmann et al., 2000*; *Figure 2C,D,G*). Strikingly, we observed that H3.3K27M expression in these arrested cells does not inhibit H3K27 trimethylation (*Figure 2L,M*). These experiments demonstrate that the H3.3K27M oncohistone is a potent inhibitor of H3K27 trimethylation, but is only effective in proliferating cells.

## Distinct K27M distributions in H3.1 and H3.3 mutant patient-derived cell lines

The genomic distribution of H3.3K27M has been previously mapped in DMG patient-derived cell lines (*Piunti et al., 2017*); however, the distribution of H3.1K27M was not known. We therefore selected a set of well-characterized DMG cell lines to characterize their epigenomes by CUT&RUN. Use of cell lines avoids the cellular heterogeneity of tumor samples that can confound chromatin profiling (*Nagaraja et al., 2019*). The monoclonal antibody we used to detect the K27M substitution reacts with the mutated residue in both H3.1 and in H3.3 histones (*Venneti et al., 2014*), and we therefore examined K27M oncohistone abundance and distribution in two H3.3K27M-carrying (SU-DIPG-XIII and SU-DIPG-XVII; 'XIII' and 'XVII') patient-derived cell lines, and in two H3.1K27M-carrying (SU-DIPG-IV and SU-DIPG-XXXVI; 'IV' and 'XXXVI') lines to compare the effects of histone subtypes.

We first confirmed that the K27M epitope was present in all four H3K27M oncohistone DMGs by Western blotting, with different amounts of oncohistone between lines (*Figure 3A*). In contrast, two high-grade glioma (HGG) cell lines (VUMC and PBT) with wild-type H3.1 and H3.3 genes show no detectable K27M signal, demonstrating the specificity of the anti-K27M antibody. The effects on abundance of H3K27me3 are not related to the amount of K27M oncohistone expressed (*Figure 3A*), but appear related to the oncohistone type. We then used the anti-K27M antibody to map distribution of the epitope across the genome in the panel of cell lines, using CUT&RUN chromatin profiling (*Skene and Henikoff, 2017*; *Skene et al., 2018*). CUT&RUN relies on binding of a specific antibody to chromosomal sites to tether a protein A-micrococcal nuclease (pA-MNase) fusion protein in samples of unfixed cells. Subsequent activation of the nuclease cleaves DNA around the binding sites with high specificity and sensitivity. Cleavage at antibody-targeted chromatin can be detected as nucleosomal fragments released from cells after nuclease activation, and we observed abundant fragments released from both H3.3K27M- and H3.1K27M-bearing cell lines (*Figure 3B*). In contrast, there is no detectable DNA released from VUMC cells, which express wild-type H3.1 and H3.3 histones, or from cells incubated with a control IgG antibody, indicating that the CUT&RUN reactions specifically cleave chromatin containing K27M oncohistones.

We focused on the results from one of each line with H3.1 (SU-DIPG-IV) or H3.3 (SU-DIPG-XIII) K27M mutations, as results for each pair of lines were very similar (see reproducibility in *Figure 3— figure supplement 1A,B* and *Figure 3—figure supplement 2*). CUT&RUN profiling of the K27M epitope showed moderate signal across the genome in both H3.1K27M and H3.3K27M lines (*Figure 3C*). This genome-wide signal is absent in cells lacking K27M oncohistones, and thus represents oncohistone that is broadly distributed. In addition, the XIII H3.3K27M line displayed 7,411 distinct peaks (*Supplementary file 1*). Notably, many of these peaks correspond to promoters, and overall these promoters are substantially enriched for K27M signal (*Figure 3E*). These peaks indicate that the H3.3K27M oncohistone is incorporated at sites of active histone turnover, superimposed upon a genome-wide background signal. To confirm that these peaks correspond to active sites, we generated profiles for the histone H3K4me2 modification – a marker of active chromatin – in H3.3K27M cells. Indeed, 44% of the K27M oncohistone peaks fall precisely at called peaks of H3K4me2 (*Supplementary file 1*). Many K27M peaks coincide with the promoters of actively transcribed genes in DMGs, including *SOX2*, *OLIG2*, *MYC* and *GFAP* (*Supplementary file 2*), and the K27M signal has a moderate correlation ($R^2 = 0.3$) with gene expression measured by RNA-seq in these cells (*Figure 3—figure supplement 3*; *Nagaraja et al., 2019*). In contrast, no H3K27M signal is observed in HGG lines with wild-type H3.3 histones (*Figure 3D,E*).

The distribution of K27M in H3.1K27M-bearing cell lines is quite different. Although H3K27M-targeted CUT&RUN released abundant amounts of nucleosome-sized DNA (*Figure 3B*) and 4–8 million fragments were sequenced and mapped, these reads are uniformly distributed across the genome for each of the two H3.1K27M-bearing cell lines (*Figure 3C*). Additionally, we found that H3K27M signal did not aggregate into defined peaks and was not enriched over promoters (*Figure 3E*) or over H3K4me2 peaks (*Supplementary file 1*). Thus, we conclude that while both H3.1 and H3.3 K27M mutations result in the incorporation of the H3K27M epitope into chromatin, the distributions

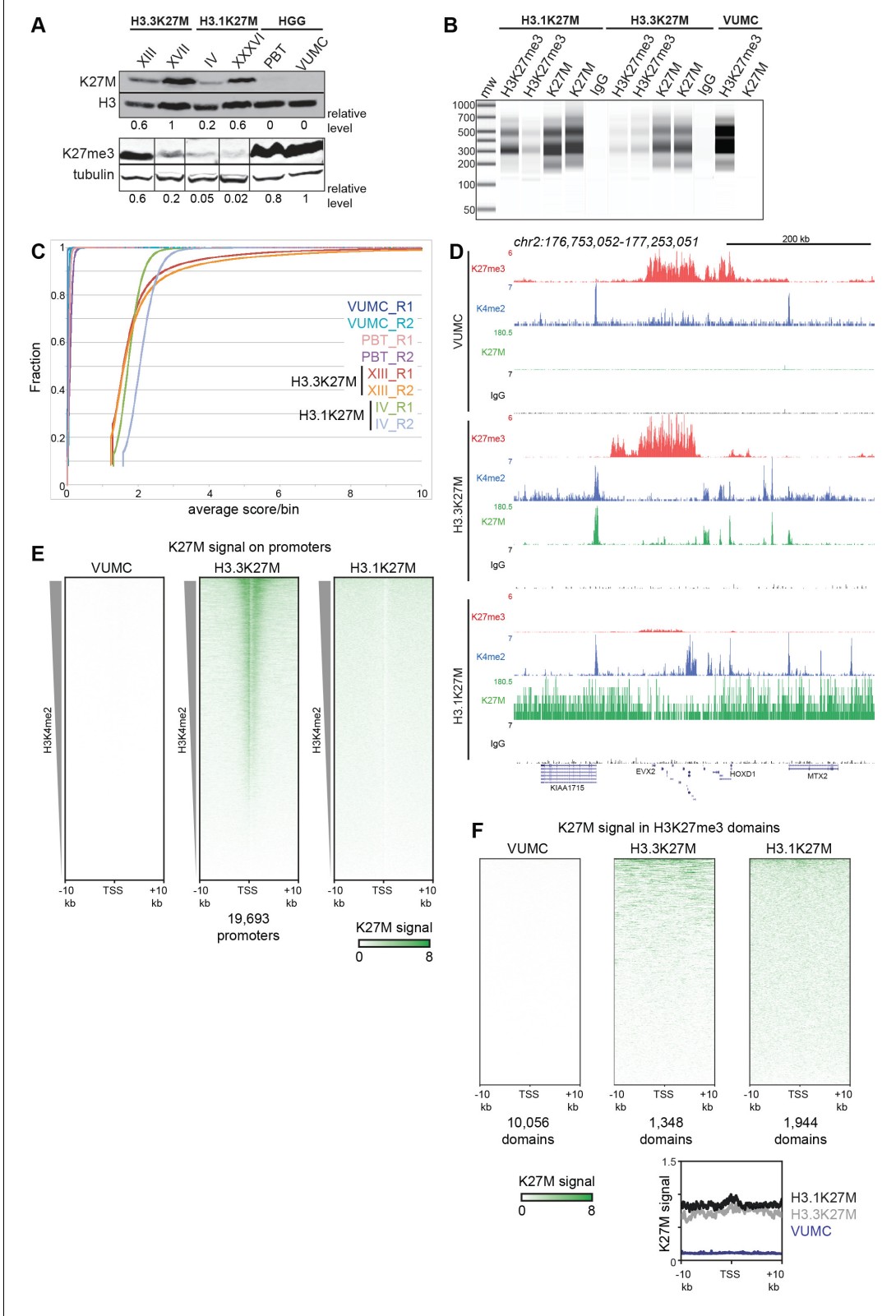

**Figure 3.** Global K27M distributions depend upon histone H3 types. (**A**) Western detection of K27M and H3K27me3 in glioma cell lines. K27M epitope amounts were normalized to histone H3 signal on the same blot, and H3K27me3 was normalized to tubulin on the same blot. The K27M histone is expressed over a range of levels in H3.1 and H3.3 mutant cell lines, but absent from VUMC and PBT gliomas. (**B**) Capillary electrophoresis of replicate CUT&RUN reactions. Antibody-targeted cleavage produce a nucleosomal ladder in amounts proportional to the amount of target epitope in samples.

*Figure 3 continued on next page*

*Figure 3 continued*

The H3 WT VUMC cell line is shown as a positive control for H3K27me3 and negative control for H3K27M. (**C**) Distribution graphs of read counts in 5 kb bins across the genome in glioma cell lines. H3.1K27M and H3.3K27M cells have moderate signal across the genome, while H3 wild-type VUMC and PBT cells have much lower counts, representing the very low non-specific background of H3K27M profiling. (**D**) Chromatin landscape of the silenced *HOXD* locus in glioma cell lines. All tracks are normalized by spike-in material to represent absolute amounts of signal. (**E**) Heat maps of H3K27M CUT&RUN signal on gene promoters in glioma cell lines. Promoters were ordered by the amount of CUT&RUN signal for H3K4me2, a mark of active transcription, in a 1 kb window around each TSS. The H3.3K27M cell line shows specific promoter enrichment of K27M in the most active promoters. (**F**) Heat maps of H3K27M across H3K27me3 domains in glioma cell lines. Domains were ordered by the amount of H3K27M signal. Both H3.1K27M and H3.3K27M cells show moderate K27M enrichment across domains. The average plot shows the enrichment of the K27M signal across the domains in each cell type.

The online version of this article includes the following figure supplement(s) for figure 3:

**Figure supplement 1.** Reproducibility of chromatin profiling experiments.
**Figure supplement 2.** Analyses of PBT-04 HGG and DIPGXVII and DIPGXXXVI glioma cell lines.
**Figure supplement 3.** Overlap between RNA-seq signal and H3K27M-enriched regions in the SU-DIPG-XIII cell line.

of the two oncohistones are consistent with replication-coupled histone deposition for both H3.1 and H3.3 oncohistones, and additional replication-independent deposition of H3.3K27M at active sites. Finally, we noted that the widespread K27M signal in both H3.1K27M and H3.3K27M lines is present in H3K27me3 domains (*Figure 3F*).

## H3K27 trimethylation is globally reduced in H3.1K27M-bearing cell lines

The K27M epitope inhibits the H3K27 methyltransferase EZH2 in vitro, in cell culture, and in H3.3K27M-bearing cell lines (*Castel et al., 2015*; *Funato and Tabar, 2018*; *Mohammad and Helin, 2017*). Indeed, western blotting for H3K27me3 shows very low levels of this histone modification in both H3.3 and H3.1 K27M mutant cells (*Figure 3A*). However, as these two mutant cell lines have very different distributions of H3K27M oncohistone, we compared their genomic profiles of H3K27me3. Since the H3K27me3 epitope is conserved between *Drosophila* and humans, we used a fixed ratio of *Drosophila* S2 cells to human cells as a spike-in control. Normalization of human read counts by *Drosophila* read counts then allows direct comparison between samples. The validity of this spike-in approach is demonstrated by the linear relationship between read counts when varying the ratio of *Drosophila* and human cells in a H3K27me3 CUT&RUN reaction (*Supplementary file 3*).

Track browsing showed high signals for H3K27me3 in H3.3K27M cells at canonical silenced domains, such as the *HOXD* locus (*Figure 3C*), as previous studies have also shown (*Piunti et al., 2017*; *Harutyunyan et al., 2019*). However, it is apparent that H3K27me3 is greatly reduced at the *HOXD* locus in H3.1K27M-bearing cells. Strikingly, it is also apparent that many methylation domains present in VUMC cells are lacking in H3.3K27M cells. To analyze differences between cell lines more thoroughly, we defined differentially methylated chromatin domains between the H3.1K27M (SU-DIPG-IV), H3.3K27M (SU-DIPG-XIII), and histone wild-type (VUMC-10) glioma lines. Unsupervised clustering of these variable H3K27me3 domains generated four major groups (*Figure 4A*, *Supplementary file 1*). The largest clusters (Clusters I-II) comprise 17,534 domains which are present in VUMC cells but absent in H3.3K27M cells, including numerous tissue-specific transcription factors. The remaining two clusters include regions where H3K27me3 is present in H3.3K27M cell lines. Cluster III includes 156 domains that are found both in VUMC and in H3.3K27M cells, and encompass genes that are typically silenced in many cell types, including those for the *CDKN2A* cell cycle inhibitor, the *WT1* tumor suppressor, and the *HOXD* transcription factors. Finally, cluster IV contains 1,380 domains that are absent in VUMC cells but present in H3.3K27M cells, including many critical regulators of developmental processes (*Supplementary file 4*). Thus, while H3.3K27M cells do have some H3K27 methylated domains, they are distinguished from other high-grade gliomas by the lack of a large number of H3K27me3 domains across the genome.

The reduction in domain numbers but not in H3K27 trimethylation levels across those domains (*Figure 4*) implies that EZH2 enzyme remains active in H3.3K27M-bearing cells. We wondered if those domains remaining in H3.3K27M DMG cells corresponds to their developmental state. Indeed, DMG cells do express transcription factors of both pluripotency and differentiated cell types, suggesting developmental similarity to both stem cells and to differentiated cells (*Filbin et al., 2018*).

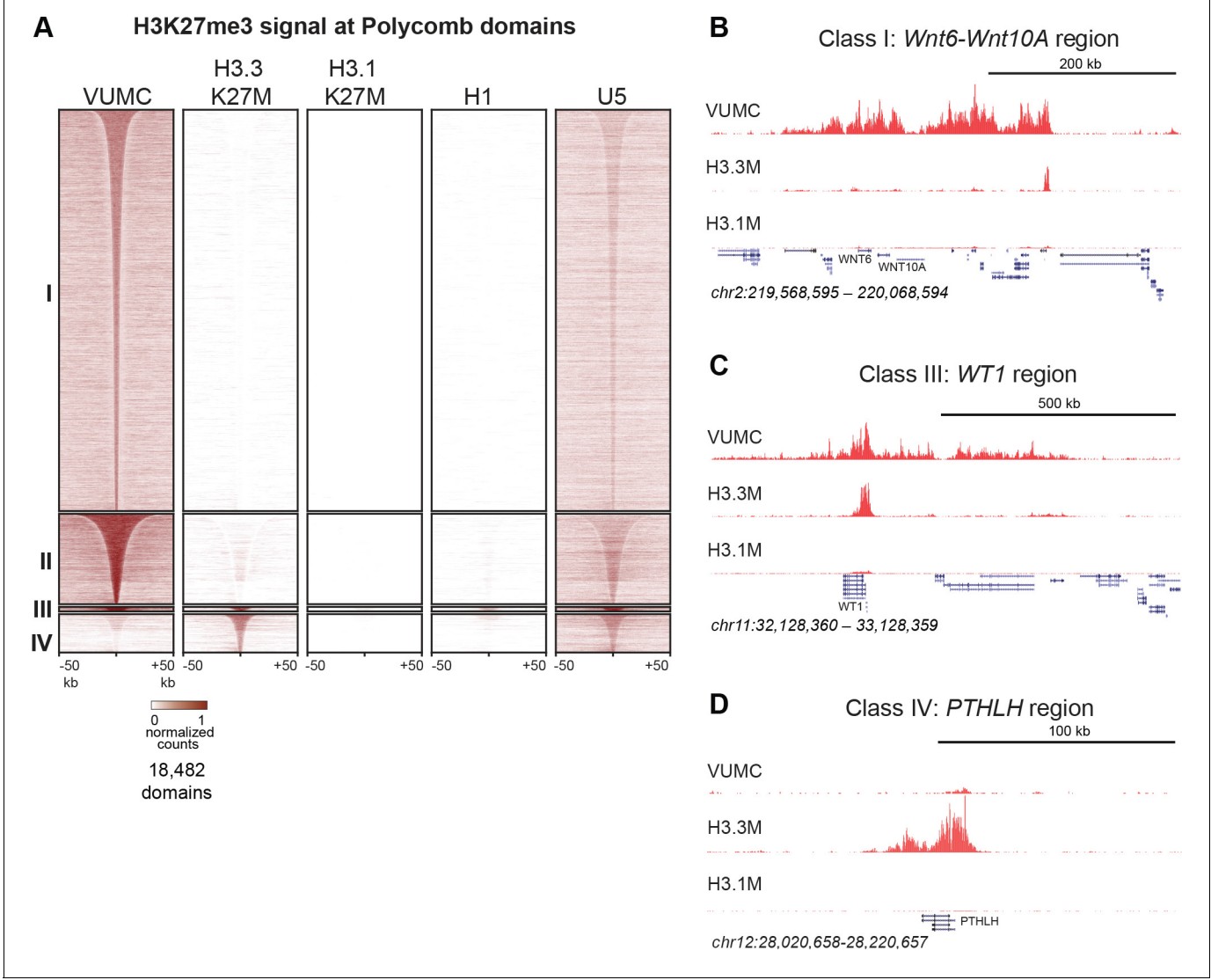

**Figure 4.** Shared and distinct H3K27me3 domains in K27M histone-expressing cells. (**A**) Heat maps of *Drosophila* normalized H3K27me3 domains in glioma, embryonic stem cell (H1) and neural stem cell (U5) cell lines. Domains that differ between VUMC, H3.3K27M and H3.1K27M cells were selected, and divided into four groups byK-means (k = 4) clustering. H3K27me3 signal across these regions was then plotted for H1 ESC and an NSC lines. Clusters I and II contain domains most enriched in VUMC and NSC lines. Cluster III contains heavily methylated domains that are present in all cell lines examined, although at low absolute levels in H3.1K27M cells. Cluster IV contains domains that are absent in VUMC cells but present to varying degrees in the other lines. (**B**) H3K27me3 landscape in a Cluster I region (present in VUMC but not in K27M histone mutant gliomas), encompassing the *WNT6* and *WNT10* oncogenes. (**C**) H3K27me3 landscape in a Cluster III region (present in VUMC and in H3.3K27M gliomas), encompassing the *WT1* tumor suppressor gene. (**D**) H3K27me3 landscape in a Cluster IV region (absent in VUMC but present in H3.3K27M gliomas), encompassing the *PTHLH* gene.

To investigate this further, we profiled H3K27 trimethylation in primitive H1 embryonic stem cells and in an untransformed neural stem cell (NSC) line from fetal forebrain (*Toledo et al., 2015*). We then examined the enrichment of histone methylation across the differential domains we defined between VUMC and K27M mutant histone gliomas. We found that 5,259 domains (21.3%) of the 24,733 domains found in U5 NSCs are found in VUMC cells, while only 1,287 (5.20%) are shared with H3.3K27M cells. In contrast, 375 domains (26.8%) of 1,401 domains in H1 ESCs are present in H3.3K27M-bearing cells. Additionally, both H3.3K27M and ES cells are distinctive in lacking H3K27 trimethylation from most Cluster I and II domains, and thus have a very limited global H3K27me3 profile.

Epigenomic changes are much more severe in H3.1K27M cells. These cells have very low bulk levels of H3K27 trimethylation, but domains can still be detected (*Supplementary file 1*), and cover many of the same regions found in H3.3K27M cells (*Figure 4A*, Clusters III and IV). These genomic patterns are consistent with other recent reports (*Castel et al., 2018*; *Nagaraja et al., 2019*). However, spike-in quantitation show that H3K27 trimethylation in these domains is reduced to 2–10% of the levels in VUMC or even in H3.3K27M cells (*Figure 3C*). Thus, the H3.3K27M and H3.1K27M oncohistones have dramatically different effects on Polycomb-regulated domains. This quantitative effect appears to be a conserved property of H3K27M mutant histones, as replication-coupled H3.2K27M oncohistones are more potent inhibitors of H3K27 trimethylation than H3.3K27M oncohistones in *C. elegans* (*Delaney et al., 2019*). We further genotyped these cell lines by targeted gene sequencing (*Kuo et al., 2020*), and identified multiple genetic alterations (*Supplementary file 5*), including biallelic deletions of the *CDKN2A* gene in the two H3.1K27M-bearing cell lines. This tumor suppressor gene is normally repressed by Polycomb silencing in normal cells, and suggests that inhibition of H3K27me3 by H3.1K27M requires secondary deletion of the *CDKN2A* locus for cell survival. In contrast, the *CDKN2A* locus is not typically altered in H3.3K27M-bearing cells (*Piunti et al., 2017*; *Mohammad and Helin, 2017*).

## DMG cells maintain PRC2 targeting

H3K27M oncohistones have been suggested to trap the PRC2 complex on chromatin and inhibit activity of the EZH2 methyltransferase complex (*Stafford et al., 2018*; *Fang et al., 2018*). To investigate the relationship between PRC2 components, H3K27me3, and K27M mutant histones in chromatin, we profiled two Polycomb proteins – SUZ12 of the PRC2 complex and the MTF2 transcription factor – in VUMC and in K27M mutant glioma lines. Similar amounts of these proteins are present in all three gliomas (*Figure 5—figure supplement 1C*). We detected thousands of sites for each subunit in each line (*Supplementary file 1*), with high concordance between biological replicates (*Figure 3—figure supplement 1*). We then mapped SUZ12 and MTF2 signal in H3.3K27M glioma cells at sites of K27M enrichment (as H3.1K27M cells only have the K27M epitope dispersed across the genome). This analysis showed no enrichment of SUZ12 or MTF2 at active promoters with the K27M signal in H3.3K27M-bearing cells (*Figure 5A*; *Figure 5—figure supplement 2A,C*). Similarly, K27M-defined peaks were not enriched for these PRC2 components (*Figure 5—figure supplement 2D*).

To examine PRC2 localization at silenced domains, we mapped SUZ12 and MTF2 signals onto the H3K27me3 clusters for each cell line (*Figure 4B*). In VUMC glioma cells, both PRC2 components are enriched in Clusters I, II, and III, where H3K27me3 is also enriched (*Figure 5A,B*). The H3.3K27M line lacks H3K27 methylation in Clusters I and II, and PRC2 components are similarly lacking. In contrast, the H3K27me3-enriched regions in Clusters III and IV are enriched for both SUZ12 and MTF2 in H3.3K27M-bearing cells (*Figure 5A,B*). Thus, in both the VUMC and H3.3K27M glioma lines, H3K27me3 domains are co-occupied by PRC2.

While H3.1K27M gliomas have dramatically reduced levels of H3K27me3 in domains, we find that those domains continue to be enriched for both SUZ12 and MTF2 proteins (Clusters III and IV, *Figure 5A*; *Figure 5—figure supplement 1*). We compared the spike-normalized levels of these proteins across Cluster III sites, which are H3K27me3 domains in all three cell lines, and found similar levels of SUZ12 and substantial levels of MTF2 in all three cell lines in these domains (*Figure 5B*). These results imply that Polycomb proteins are correctly targeted in both H3.1K27M and H3.3K27M cells, but PRC2 must be strongly inhibited in H3.1K27M cells to account for their low H3K27me3 signal.

## Discussion

Replication-coupled and -independent mechanisms of H3 variant deposition were first demonstrated almost two decades ago (*Ahmad and Henikoff, 2002*) but the roles of these different deposition mechanisms in H3.1-and H3.3-mutant gliomagenesis are not known. We use a panel of high-grade glioma and DMG cell lines to profile the genome-wide localizations of K27M oncohistones. Previous reports for the localization of H3.1K27M have been contradictory, identifying accumulation at active promoters (*Piunti et al., 2017*) or more broadly throughout the genome (*Nagaraja et al., 2019*). Our results confirm that H3.1K27M is deposited throughout the genome, consistent with replication-coupled deposition of H3.1, while H3.3K27M accumulates primarily at sites of histone turnover but

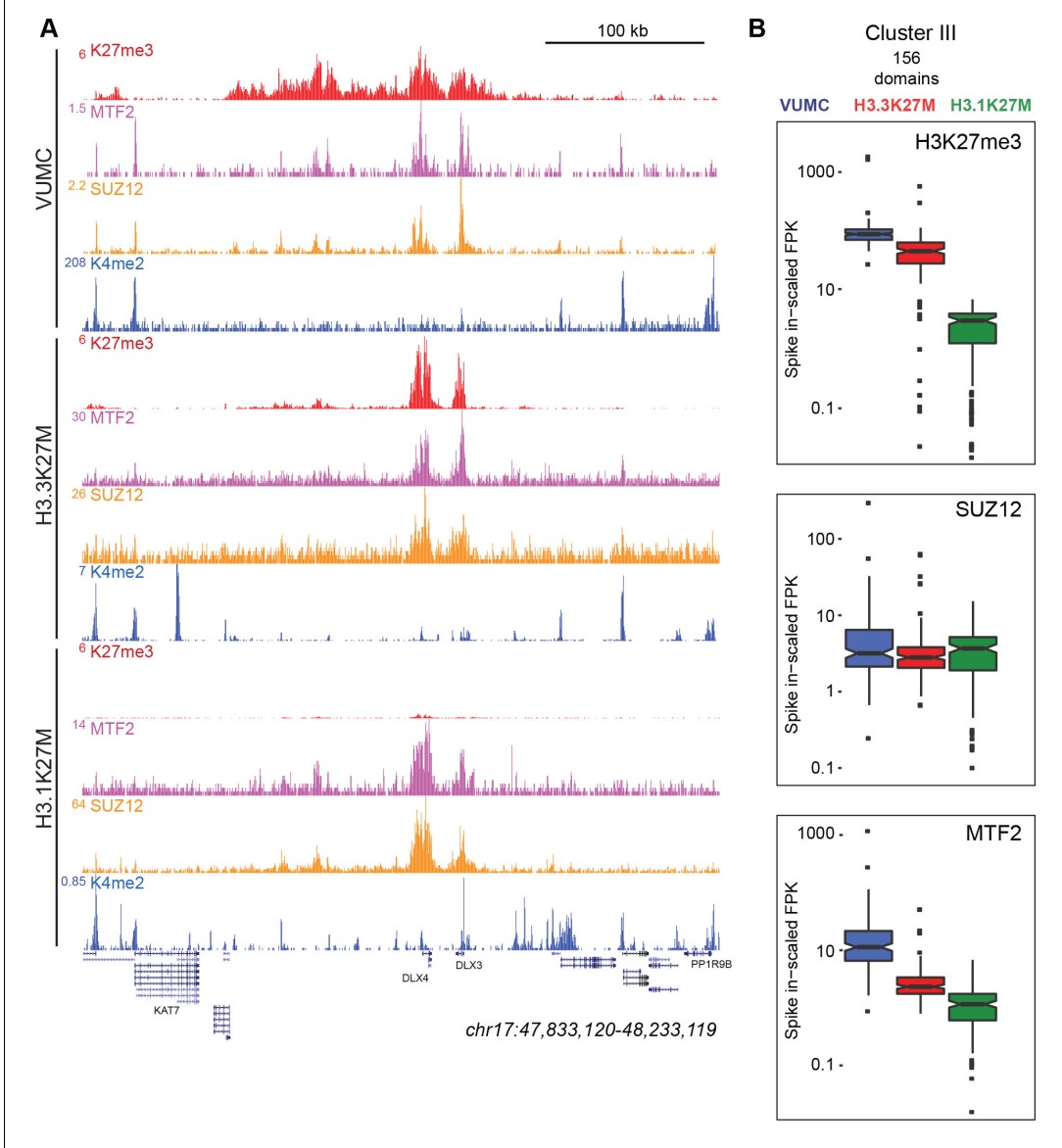

**Figure 5.** K27M histone-expressing cells maintain PRC2 binding. (**A**) Chromatin landscape around the *DLX3* gene for H3K27me3, H3K4me2, and Polycomb components in glioma cell lines. All three cell lines display binding of Polycomb components within the H3K27me3 domain (**B**) Bar-and-whisker plots of H3K27me3, SUZ12, and MTF2 signals in shared Cluster III domains in VUMC, H3.3K27M (XIII), and H3.1K27M (IV) cell lines. H3K27me3 signal is similar between VUMC and H3.3K27M lines, but low in H3.1K27M cells. In contrast, the PRC2 subunit SUZ12 is similarly enriched in all three cell lines at these shared domains, while MTF2 enrichment varies between lines.

The online version of this article includes the following figure supplement(s) for figure 5:

**Figure supplement 1.** Enrichment of PRC2 in H3K27me3 domains.

**Figure supplement 2.** Enrichment of PRC2 components at sites of active histone turnover.

also at low levels genome-wide, probably through replication-coupled deposition. Furthermore, quantitative profiling demonstrates that H3K27me3 is much lower in H3.1K27M- than in H3.3K27M-bearing cells, although neither the H3.1K27M nor H3.3K27M oncohistones appear to trap or prevent PRC2 binding to chromatin.

Competing models have suggested that H3K27M oncohistones sequester (*Fang et al., 2018*) or poison (*Lee et al., 2019*) PRC2. Further, it has been suggested that these effects occur either on chromatin or in solution (*Nagaraja et al., 2019*). However, PRC2 components in DMG cells do not coincide with the bulk of K27M oncohistones in these cells. Taken together, our human and fly

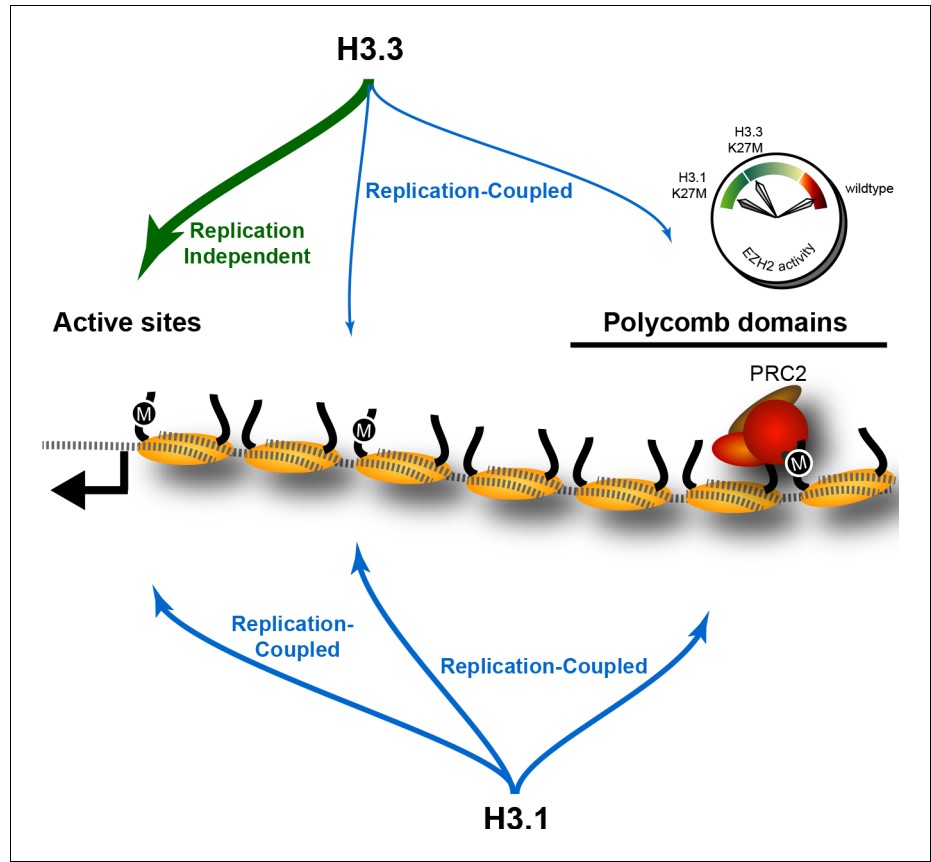

**Figure 6.** The effect of K27M histones is determined by histone deposition pathways and the distribution of PRC2 complexes on chromatin. Nucleosomes (yellow) are assembled from new histones by replication-coupled (blue arrows) and replication-independent (green arrow) pathways. Most H3.3 histones are bound by variant-specific chaperones and incorporated at active sites and promoters by replication-independent pathways, but a small amount of H3.3 histones are incorporated during DNA replication throughout the genome. The H3.1 histone is exclusively used for replication-coupled assembly, and so deposited throughout the genome at higher levels. Thus, H3.1K27M mutants incorporate high levels of the K27 ('M') epitope in polycomb domains, where it binds and inhibits the histone methyltransferase activity of the PRC2 complex. In contrast, most K27M epitope in H3.3K27M mutants is incorporated at active sites far from polycomb domains, resulting in weaker inhibition of PRC2 in these cells.

results provide a coherent model for inhibition by H3K27M oncohistones (*Figure 6*). We have demonstrated that the K27M epitope only inhibits H3K27 trimethylation on chromatin in vivo, supporting the idea that these oncohistones inhibit chromatin-bound PRC2 complexes. A possible mechanism comes from recent reports showing that EZH2 methylates itself, and that this automethylation is required for full catalytic activity (*Lee et al., 2019*; *Wang et al., 2019*). These studies also showed that H3K27M blocks EZH2 automethylation, and might contribute to the dominant effect of H3K27M mutations.

Our results imply that H3K27M oncohistones inhibit the EZH2 methyltransferase at its chromatin-binding sites within pPolycomb-regulated domains. A critical time for maintaining epigenomic patterns is during S phase of the cell cycle, when histone modifications must be established on newly assembled nucleosomes behind the replication fork (*Alabert et al., 2015*). It is striking that H3K27M oncohistones inhibit H3K27 trimethylation only in proliferating cells (*Chan et al., 2013*; *Nagaraja et al., 2019*). Existing H3K27me3 levels generally remain stable in the absence of DNA replication (*Jadhav et al., 2020*), thus EZH2 activity is only needed in cycling cells. Perhaps new H3K27M oncohistones deposited behind the replication fork inhibit the EZH2 methyltransferase, thereby blocking re-establishment of H3K27me3 domains. In this model, only H3K27M oncohistones

that undergo replication-coupled deposition near the sites of PRC2 recruitment are toxic to EZH2 activity. As H3.1K27M oncohistones are deposited throughout the genome, PRC2 poisoning in these cells eliminates H3K27 trimethylation genome-wide. In contrast, the less potent effect of H3.3K27M oncohistones on H3K27 trimethylation in DMG cells may be due to the smaller fraction of H3.3K27M oncohistones that undergo replication-coupled deposition, as most of this oncohistone is sequestered at active promoters far from H3K27me3 domains. The finding that human histone H3.3 localize over replication foci in early-S-phase cells (*Ray-Gallet et al., 2011*; *Clément et al., 2018*) is consistent with our model.

That only H3K27M oncohistones deposited during DNA replication are toxic to EZH2 activity also explains why particular secondary mutations are associated with either H3.1K27M- or H3.3K27M-bearing gliomas. For example, H3.3K27M but not H3.1K27M gliomas have a significantly increased frequency of *ATRX* mutations (*Khuong-Quang et al., 2012*; *Mackay et al., 2017*). ATRX is a chromatin remodeler involved in alternative telomere lengthening pathways in some cells (*Heaphy et al., 2011*), but also mediates replication-independent histone deposition (*Goldberg et al., 2010*). Thus, we expect that *ATRX* mutations may enhance the amount of H3.3K27M oncohistones deposited during DNA replication, thereby increasing inhibitory effects on H3K27 trimethylation. Similarly, *CDKN2A* loss in H3.1K27M-bearing cells suggests that this oncohistone requires different secondary mutations to compensate for the loss of Polycomb silencing at tumor suppressor genes. Such a dependency would explain why H3.1K27M mutations are more rare than H3.3 ones. A therapeutic corollary is that H3.1 and H3.3 oncohistones may confer distinct cellular sensitivities to chromatin modification inhibitors, either by affecting chromatin silencing or by affecting histone deposition pathways, thereby reshaping epigenomic landscapes.

## Materials and methods

### Biological materials
#### Cell lines
Patient-derived K27M SU-DIPG-IV (IV, H3.1K27M), SU-DIPG-XXXVI (XXXVI, H3.1K27M), SU-DIPG-XIII (XIII, H3.3K27M) and SU-DIPG-XVII (XVII, H3.3K27M) cell lines were generously provided by the laboratory of M. Monje (Stanford University) and have been previously described (*Grasso et al., 2015*). The high-grade glioma (HGG) cell line with wild-type histone genes VUMC-DIPG-10 (VUMC) (*Meel et al., 2017*) was obtained through a materials transfer agreement with Esther Hulleman (VU University Medical Center, Amsterdam, Netherlands), and generation of PBT-04 was reported previously (PBT; *Brabetz et al., 2018*). The H1 ESC line was obtained from WiCell (Madison WI). The U5 neural stem cell (NSC) line (*Toledo et al., 2015*) was obtained from PJ Paddison (FHCRC, Seattle, WA). *Drosophila* S2 cells were obtained from ThermoFisher. The patient-derived glioma cell lines harboring H3K27M mutations (SU-DIPG-IV, SU-DIPG-XIII, SU-DIPG-XVII, SU-DIPG-XXXVI) had targeted sequencing to confirm H3 mutational status. PBT-04 was generated by the Olson Laboratory and maintained as described previously (*Brabetz et al., 2018*). VUMC-10 was generously provided directly by Dr. E. Huelleman. Mycoplasma testing was performed every 3 months with the MycoProbe mycoplasma detection kit from R and D systems (Minneapolis, MN). None of the cell lines used in this study are found on the misidentified cell lines list from the International Cell Line Authentication Committee.

#### Fly lines and crosses
The *GMR-GAL4-D* driver (*Ahmad and Henikoff, 2001b*), *UASp-H3.3K27M* (*Ahmad and Spens, 2019*), and *GMR-p21* (*Ollmann et al., 2000*) lines were used. All crosses were performed at 25°C. Crawling 3rd instar larvae were selected, and eye discs were dissected and fixed in 4% paraformaldehyde/PBST (PBS with 0.1% triton-X100). Fixed tissues were blocked with 10% goat serum/PBST, and incubated with primary antiserum at 4° overnight, and with fluorescently labeled secondary antibodies (1:200 dilution, Jackson ImmunoResearch). All tissues were stained with 0.5 µg/mL DAPI/PBS, mounted in 80% glycerol on slides, and imaged by epifluorescence on an EVOS FL Auto 2 inverted microscope (Thermo Fisher Scientific) with a 10X or 20X objective. Pseudo-colored images were adjusted and composited in Adobe Photoshop and Adobe Illustrator. H3K27me3 signal in the

anterior and posterior portions of eye discs was measured as the mean value in a 100 pixel x 100 pixel box using Photoshop.

## Plasmid constructs

We built plasmids by Gibson assembly for constitutive expression of histones and K27M mutant histones tagged at their C-termini with 2XFLAG epitope tags using the *Copia* promoter from the plasmid pCoPURO (Addgene 17533), and *Drosophila* H3 and H3.3 histones from previously published constructs (*Ahmad and Henikoff, 2002*). Site-directed mutagenesis was used to introduce K27M substitutions into the H3 and H3.3 genes. To make histone tail constructs, we made a Copia-H3-RFP fusion construct and then used site-directed mutagenesis to delete the histone fold domain. All plasmids were confirmed by Sanger sequencing of the fusion gene. The following plasmids are used here: pCoH3_FLAG, pCoH3K27M_FLAG, pCoH3.3_FLAG, pCoH3.3K27M_FLAG, pCoH3tail_RFP, and pCoH3K27Mtail_RFP.

## Cell culture

Human cells were grown in NeuroCult medium (StemCell Technologies, Vancouver, BC) supplemented with human-EGF at 20 ng/mL and human-bFGF at 20 ng/mL supplemented with penicillin/streptomycin. Cells were passaged with Accutase for dissociation. *Drosophila* S2 cells were grown to log phase in HYQ-SFX insect medium (ThermoFisher) supplemented with 18 mM L-Glutamine and harvested by scraping. *Drosophila* cells were transiently lipofected with plasmid constructs using Fugene HD (Promega), grown for 2 days, then spun onto glass slides in a Cytospin centrifuge (Thermo), fixed, and immunostained as described (*Ahmad and Henikoff, 2001a*). Nuclei were photographed using an EVOS FL Auto two inverted microscope (ThermoFisher) with a 20X lens. Two transfections were performed for each experiment. Nuclei were scored for transfection by the construct marker (RFP or FLAG), and the mean signal of H3K27me3 staining was measured for 50 untransfected and 50 transfected nuclei in the DAPI-stained nucleus using Photoshop CS6 Extended for each construct. Signals were corrected for background on slides and then divided by the brightest H3K27me3-stained nucleus to normalize between slides and images.

## Western blotting

For histone western blots, $10^6$ cells were pelleted, washed once with PBS and resuspended in 200 μL standard protein sample buffer to make whole cell extracts. Samples were vortexed, boiled for 5 min, then cooled to room temperature. Benzonase (1 μL) was added and samples were incubated at room temperature for 5 min before freezing for further use. For SUZ12 and MTF2, chromatin fractions were isolated using acid extraction. Samples were run on 4–20% Tris-Glycine polyacrylamide gels (Invitrogen), transferred to nitrocellulose membrane and 1:1000 dilutions of primary and secondary antibodies were used for blotting. SUZ12 was probed first and the same blot was stripped with Stripping Buffer (LiCor, Lincoln, NE) and then probed with anti-MTF2. Secondary goat anti-mouse IRDye800CW and goat anti-rabbit IRDye680LT (LI-COR, Lincoln, NE) were used and quantification was performed using the ImageJ software, accounting for local background and with internal loading control.

## CUT&RUN chromatin profiling

CUT&RUN was performed as described (*Skene et al., 2018*). All antibodies were used at 1:100 dilutions, except anti-SUZ12 and anti-MTF2, which were used at 1:50. Cells were counted using a ViCell (ThermoFisher), and spike-in CUT&RUN was performed with a 1:20 ratio of *Drosophila* S2 cells to human cells (50,000 S2 cells to 1,000,000 human cells) in each reaction.

## Library preparation and sequencing

Extracted DNA was subjected to the KAPA Hyper-prep library preparation kit protocol (Roche, Inc) and amplified as previously described (*Skene and Henikoff, 2017*), with the modification that the end-repair reaction and poly-A tailing reactions were performed at 60˚C to preserve small fragments, as described (*Liu et al., 2018*). Sequencing reads were mapped to the human hg19 genome build and the genome build for the relevant spike-in for each dataset *Drosophila* dm6 for H3K27me3, *Saccharomyces cerevisiae* R64-1-1 for K27M, or *E. coli* Ensembl genome build for (SUZ12 and MTF2)

using Bowtie2 (*Langmead and Salzberg, 2012*), and paired-end fragment bed files and spike-normalized bedgraphs generated using bedtools (*Quinlan and Hall, 2010*). Spike-normalization factors were calculated by dividing 10,000 by the number of spike-in reads mapped, and for each sample the bedgraph signal was multiplied by the corresponding factor.

## Data analysis

Correlation heatmaps were generated in R (https://www.rproject.org), using normalized fragment counts mapping to 10 kb windows spanning the hg19 genome. Peaks were called using SEACR (*Meers et al., 2019*). For datasets for which biological replicates were available, peak lists from two replicates were merged using the bedops merge (-m) utility (*Neph et al., 2012*). For genome-wide assessment of CUT&RUN signal, we used deepTools to sum basepair counts in 5 kb bins, and plotted the ranked counts in MS Excel. Differential H3K27me3 regions used for K-means clustering were generated as follows: (1) We used the bedops partition (-p) utility to determine unique overlapping segments between merged H3K27me3 peak lists from VUMC, XIII, and IV datasets; (2) From the partitioned peak list, for any two partitioned regions that were directly adjacent to one another, we filtered out any that were less than 10 kb in length, in order to select only the 'dominant' segment in each region of partitioning overlaps, and to avoid doubly mapping adjacent partitioned regions; (3) For all other free-standing regions, we filtered out any that were less than 3 kb in length. This resulted in 18,482 regions used for clustering. K-means clustering was carried out using the 'kmeans' utility in R. Four clusters were selected as the optimal cluster solution based on selecting the 'knee' of the curve in a plot of the number of clusters vs. within-cluster distance, tested across all possible solutions between 2 and 15 clusters. Overlaps between cluster regions and gene promoters, and mapping of H3K27me3, SUZ12 and MTF2 fragments to cluster III regions in *Figure 5B*, were ascertained using the bedtools intersect utility (*Quinlan and Hall, 2010*). Fragments mapped to cluster III in *Figure 5B* were first scaled by fragments per kilobase (1000/length of region), and then by a scaling constant that is inversely proportional to the number of spike in reads mapped (VUMC-H3K27me3: 0.786, SUDIPG-XIII-H3K27me3: 0.292, SUDIPG-IV-H3K27me3: 0.046, VUMC-SUZ12: 0.104, SUDIPG-XIII-SUZ12: 0.385, SUDIPG-IV-SUZ12: 0.185, VUMC-MTF2: 0.244, SUDIPG-XIII-MTF2: 0.333, SUDIPG-IV-MTF2: 0.084). Detailed positional information for all merged peak calls used in this manuscript are found in *Supplementary file 1*. Signal heatmaps, gene plots, and average plots were generated using deepTools (*Ramírez et al., 2014*). RNA-seq data for SU-DIPG-XIII was obtained from GSM2471870 (https://www.ncbi.nlm.nih.gov/geo/query/acc.cgi?acc=GSM2471870). For comparison of RNA-seq counts with H3K27M in SUDIPG-XIII, reads were mapped in a 10 kb window surrounding hg19 TSSs. Datasets were visualized using the UCSC Genome Browser. Boxplots were generated with the web application BoxPlotR (http://shiny.chemgrid.org/boxplotr/).

## Acknowledgements

We thank Ekaterina Babaeva, Shelli Morris and Matthew Biery for technical assistance. We thank Michelle Monje (Standford University) for providing SU-DIPG-IV, SU-DIPG-XIII, SU-DIPG-XVII and SU-DIPV-XXXVI cell lines, Esther Hulleman VU (University Medical Center, Amsterdam) for providing the VUMC-DIPG-10 line, Eliza Small (ThermoFisher) for the MTF2 antibody, Christine Codomo for library preparation, and Srinivas Ramachandran for helpful discussion and analysis. JFS is supported by a Damon Runyon-Sohn Foundation Fellowship, received support from NIH NCI Training Grant T32 CA009351 and an Alex's Lemonade Stand Young Investigator Award. This work was funded by the HHMI (SH) and by the NIH (R01GM108699, KA).

## Additional information

### Funding

| Funder | Grant reference number | Author |
| --- | --- | --- |
| Howard Hughes Medical Institute | | Steven Henikoff |
| National Institutes of Health | R01GM108699 | Kami Ahmad |

| Alex's Lemonade Stand Foundation for Childhood Cancer | | Jay F Sarthy |
| Damon Runyon Cancer Research Foundation | | Jay F Sarthy |
| National Institutes of Health | T32 CA009351 | Jay F Sarthy |

The funders had no role in study design, data collection and interpretation, or the decision to submit the work for publication.

## Author contributions

Jay F Sarthy, Conceptualization, Investigation, Methodology, Writing - original draft, Writing - review and editing; Michael P Meers, Conceptualization, Software, Formal analysis, Investigation, Methodology, Writing - original draft, Writing - review and editing; Derek H Janssens, Investigation, Methodology; Jorja G Henikoff, Software, Formal analysis; Heather Feldman, Patrick J Paddison, Christina M Lockwood, Nicholas A Vitanza, Resources; James M Olson, Resources, Supervision; Kami Ahmad, Conceptualization, Resources, Funding acquisition, Investigation, Methodology, Writing - original draft, Writing - review and editing; Steven Henikoff, Conceptualization, Resources, Supervision, Funding acquisition, Investigation, Methodology, Writing - original draft, Writing - review and editing

## Author ORCIDs

Jay F Sarthy (ID) https://orcid.org/0000-0001-5244-7865
Michael P Meers (ID) https://orcid.org/0000-0003-3438-3938
Steven Henikoff (ID) https://orcid.org/0000-0002-7621-8685

## Decision letter and Author response

Decision letter https://doi.org/10.7554/eLife.61090.sa1
Author response https://doi.org/10.7554/eLife.61090.sa2

# Additional files

## Supplementary files

• Supplementary file 1. MSExcel spreadsheets of all called peaks and domains from profiling experiments.

• Supplementary file 2. Gene IDs for H3K27M-enriched peaks from H3K27M performed in H3.3K27M-mutant DMG cell lines.

• Supplementary file 3. Sequencing read counts for each profiling experiment.

• Supplementary file 4. Gene Ontology term analysis for genes overlapped by differential H3K27me3 domains in Cluster IV.

• Supplementary file 5. Oncoplex sequencing results of the H3K27M-positive high grade glioma cell lines.

• Transparent reporting form

## Data availability

Sequencing data have been deposited in GEO under accession code GSE118099.

The following dataset was generated:

| Author(s) | Year | Dataset title | Dataset URL | Database and Identifier |
|---|---|---|---|---|
| Sarthy JF, Meers MP, Ferguson E, Janssens DH, Vitanza NA, Ahmad K, Olson JM, Henikoff S | 2020 | Cell-of-origin and Developmental Trajectories Cooperate to Determine Chromatin Landscapes in Histone-Mutant Diffuse Midline Gliomas | http://www.ncbi.nlm.nih.gov/geo/query/acc.cgi?acc=GSE118099 | NCBI Gene Expression Omnibus, GSE118099 |

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
