## [Decision Letter]

**Decision letter after peer review:**

[Editors’ note: the authors submitted for reconsideration following the decision after peer review. What follows is the decision letter after the first round of review.]

Thank you for submitting your work entitled "Histone deposition pathways determine the chromatin landscapes of H3.1 and H3.3 K27M oncohistones" for consideration by *eLife*. Your article has been reviewed by three peer reviewers, including Jerry L Workman as the Reviewing Editor and Reviewer #1, and the evaluation has been overseen by a Reviewing Editor and a Senior Editor. The following individuals involved in review of your submission have agreed to reveal their identity: David M. Gilbert (Reviewer #2).

Our decision has been reached after consultation between the reviewers. Based on these discussions and the individual reviews below, we regret to inform you that your work will not be considered further for publication in *eLife*.

The reviewers felt that the model proposed is interesting, however, the data supporting the conclusions are not sufficient at this time. This is detailed in the attached reviews. The reviewers also felt that you should be encouraged to submit a stronger manuscript on this topic to *eLife* in the future.

Reviewer #1:

This is a straightforward study that characterizes the effects of K3.1 vis H3.3. K27M oncohistone mutations on poly comb occupancy and methylation in mammalian cells and flies. While the results are a bit predictable. H3.1K27M is wider spread and more toxic than H3.3K27M the work does address some important mechanistic details. (1) inhibition of EZH2 does not occur in solution. (2) inhibition appears to be in cis and not in trans. (3) H3K27M does not appear to trap EZH2 or prevent its binding. These are important contributions.

1) the following statement seems contradictory with the overall conclusions. please clarify.

"This analysis showed no significant binding of either Polycomb component at active promoters, which have the K27M epitope in H3.3K27M cells (Figure 5A, Figure 5—figure supplement 2A,C) or at K27M-defined peaks in the VUMC cell line (Figure 5—figure supplement 2AB,D )."

2) Is EZH2 automethylation inhibited by H3K27M?

Reviewer #2:

In this manuscript the authors perform experiments in *Drosophila* and human cell lines to investigate they hypothesis that H3K27M mutation poison PRC2-mediated H3K27 tri-methylation. They provide evidence that, in *Drosophila*, H3.2 inhibition of K27 methylation requires incorporation of H3.2 into nucleosomes and in a separate series of experiments they show that over-expressed H3.3 inhibition of K27 methylation requires proliferation. They then perform Cut&RUN experiments for H3K27M, H3K4me2 and H3K27me3 in several human glioblastoma cell lines. These experiments show different patterns of K27me3 inhibition in H3.3K27M vs. H3.1 K27M harboring cell lines. The paper is generally well-written and easy to follow and the experiments themselves show high quality data. My biggest problem with the manuscript is the incompleteness and incongruency of some of the approaches being cobbled together into universal conclusions that are not entirely warranted. The human results seem much more consistent with replication-dependent vs. independent effects so it is possible that there are artifacts with the fly data (see below) or that flies and humans are not the same. I am also not clear on what are the truly novel findings here. The proliferation-dependence of H3.3 mutant incorporation is intriguing but the one experiment is rather preliminary. It has been known that the H3.1 and 3.3 -associated gliomas have different transcriptomes and epigenomes and that is not really a surprise. It has been shown that the K27M epitope inhibits EZH2 in vitro, in cell culture, and in H3.3K27M-bearing cell lines

Essential revisions:

1) The suggestion that H3.3 inhibition of K27me3 requires proliferation is intriguing since the majority of H3.3. is incorporated into histones in a replication-independent manner but it is too preliminary. The authors need to rule out that the effect is not due to the over-expression of H3.3 is driving its incorporation via the CAF1 pathway in their experiments and they need to demonstrate that it is replication directly that is the dependency and not some indirect effect of p21 expression which surely changes many things. The authors rely strongly on this result in their model in the Discussion, which they present as conserved fly to human, but this experiment in flies is not convincing and whether the same is true in human is not addressed. Then in light of the human cell results, where the patterns of change in H3K27me3 are more consistent with replication independent and dependent incorporation of the H3.3 variant in hot areas while the 3.1 variant is widespread; I don't understand why they insist that the fly results be incorporated into the model of what they see in the human cell lines.

2) The authors look at only one variant in each *Drosophila* approach, but draw universal conclusions.

3) Subsection “G cells maintain PRC2 targeting” – how are we to judge what number of domains "highlights developmental similarity" – we need statistics and negative and positive controls to conclude this. Otherwise all we know is that some are similar and some are different as I suspect would be true in any comparison. Also, for the phrase "akin to stem cells" do the authors refer to embryonic or neural stem cells. If both, then negative controls are very necessary. The total numbers of domains are very different between these two types of stem cells so I'm not even sure that statistics (not discussed) would be helpful but they at least should be stated.

Reviewer #3:

In this manuscript, the authors investigate how oncohistones carrying K27M mutations inhibit K27me3, focusing on the role of histone variant subtypes. This work is of high quality and the findings are interesting to the chromatin field. Several of the results in this manuscript cooperates previous work on oncohistones, and the novelty lies in the finding that H3K27M oncohistones only reduce K27me3 level in proliferating cells and the quantitative comparison of H3.1K27M and H3.3K27M cells. The proposed model is interesting, but needs further support. Further dissection of features that distinguish sites that maintain/loose K27me3 in H3.3K27M cells would also strengthen the manuscript. With these revisions the work could be recommended for publication in *eLife*.

Essential revisions:

- The authors propose a model that entails replication-coupled incorporation of H3.3 at a low level. However, this work does not provide evidence to support this claim (it does not investigate this point directly) and evidence for this in the literature is weak – Drane et al. show that CAF-1 binds H3.3 in DAXX KO cells and Ray-Gallet et al., explores HIRA dependent GAP filling following replication when CAF-1 function is impaired.

The experiment in Figure 2 shows convincingly that replication is required for H3.3K27M to exert its negative effect on PRC2 and K27me3 levels. However, this could be explained without invoking replication-coupled deposition of H3.3K27M:

- Existing K27me3 levels generally remain stable in absence of DNA replication despite reduced PRC2 function (Jadhav et al., 2020). The major requirement for PRC2 function arises with incorporation of large numbers of unmodified histones, likely explaining why H3.3K27M interferes with K27me3 maintenance only in dividing cells. Moreover, H3.3 incorporation is also expected to increase after replication – via a replication-independent mechanism but due to replication-dependent doubling of the genome.

The authors should either provide evidence to support H3.3 incorporation genome-wide through a replication-coupled mechanism or consider to revise the model. They should also investigate the possibility that replication-dependent incorporation of new histones is required to see any effect of PRC2 inhibition in their setup – e.g. would they see loss of H3K27me3 in the *Drosophila* model system in Figure 2 if EZH2 was inhibited in the presence of p21 expression?

- The authors show that K27me3 patterns in H3.3K27M cells is more similar to H1 ESCs than VUMC glioma cells. Is this reflecting that the cells are blocked at an earlier developmental stage and if so do gene expression analysis support this notion? what would be the alternative explanation for a specific lack of K27me3 in cluster I and II.

Is there a difference in H3.3K27M occupancy (histone exchange) across cluster I-IV that may explain why some clusters are more prone to K27me3 loss?

[Editors’ note: further revisions were suggested prior to acceptance, as described below.]

Thank you for resubmitting your work entitled "Histone deposition pathways determine the chromatin landscapes of H3.1 and H3.3 K27M oncohistones" for further consideration by *eLife*. Your revised article has been evaluated by Jessica Tyler (Senior Editor) and Jerry Workman (Reviewing Editor}.

Please note that reviewers 2 and 3 were not reviewers on the previously submitted version of the manuscript. However, they had access to the previous reviews and your responses, which they judged were appropriate.

The manuscript has been improved, but there are some remaining issues that need to be addressed before acceptance, as outlined below:

In this work, the authors use both *Drosophila* cells/tissues and human patient-derived DMG cells to study two oncohistones H3.1K27M and H3.3K27M. The main conclusions are: (1) Inhibition of oncohistone on K27me3 only happens in cis when mutant histones are incorporated into chromatin and only in dividing cells. (2) They showed that the distinct deposition modes of H3.1 and H3.3 determine their genome distribution with H3.1K27M showing a more global distribution while H3.3K27M having more localized enrichment. (3) Neither H3.1K27M nor H3.3K27M oncohistone interferes with PRC2 binding.

The experimental strategies combined individual cell assay such as immunostaining and genomic analysis such as CUT&RUN, the later gave out clean results with high S/N ratio. The results are presented in a logic manner and the main conclusions are made carefully (except the third one, see Essential revision #4 below). However, for several places, the presentation of the results needs revision or improvement. Overall, this work should be appropriate for *eLife* after revisions.

Essential revisions:

1) Figure 1: In transfected *Drosophila* S2, if cells are not allowed to progress "2-3 cell cycles", do they lose or maintain H3K27me3? This would be good to include as a control, and for the conclusion cell cycle progression is required.

2) Figure 2 uses eye imaginal discs to study the effect of oncohistones. This is a great design as cells either exit cell cycle or undergo active mitosis can be visualized all together. A few suggestions on this figure: (1) Only H3.3K27M is studied here, why leaving H3K27M out for this experiment? (2) Figure 2A-B need to improve image resolution, it is quite blurry. (3) The H3S10 phosphorylation immunostaining signal is unclear, it is hard to tell the synchronously dividing cells right posterior to the MF. It would be better to show a better resolution image with this channel separate in gray scale. (4) In Figure 2 legend, it is said that "at least 10 eye discs were examined for each genotype". In the figure, only one eye or eye disc was shown. First, N=10 is not a large sample size; second, quantification on them should be shown or mentioned in the text, such as what percentage of the samples show what phenotype, etc.

3) Figure 3E: Further analysis on H3.3K27M-enriched promoters should be informative to understand the molecular mechanisms of this oncohistone.

4) For this conclusion-- "Neither oncohistone interferes with PRC2 binding." The authors examined binding of SUZ12 and MTF2 in Figure 5, however, the key PRC2 component that should be studied is EZH2, the methyl-transferase. It would be very informative to examine EZH2's binding here.

---

## [Author Response]

[Editors’ note: the authors resubmitted a revised version of the paper for consideration. What follows is the authors’ response to the first round of review.]

Reviewer #1:This is a straightforward study that characterizes the effects of K3.1 vis H3.3. K27M oncohistone mutations on poly comb occupancy and methylation in mammalian cells and flies. While the results are a bit predictable. H3.1K27M is wider spread and more toxic than H3.3K27M the work does address some important mechanistic details. (1) inhibition of EZH2 does not occur in solution. (2) inhibition appears to be in cis and not in trans. (3) H3K27M does not appear to trap EZH2 or prevent its binding. These are important contributions.1) the following statement seems contradictory with the overall conclusions. please clarify."This analysis showed no significant binding of either Polycomb component at active promoters, which have the K27M epitope in H3.3K27M cells (Figure 5A, Figure 5—figure supplement 2A,C) or at K27M-defined peaks in the VUMC cell line (Figure 5—figure supplement 2B,D )."

We have reworded this sentence in subsection “H3K27 trimethylation is globally reduced in H3.1K27M-bearing cell lines” for clarity as follows: “This analysis showed no binding of SUZ12 or MTF2 at active promoters with the K27M epitope in H3.3K27Mbearing cells above the background observed in VUMC cells, which lack the K27M epitope (Figure 5A, Figure 5—figure supplement 2A,C). Similarly, no enrichment above background was apparent at K27M-defined peaks (Figure 5—figure supplement 2B,D)”. Furthermore, we previously made the observation in the Results section but did not fully interpret it, and we have added a sentence to the Discussion section pointing out that this result is not consistent with existing models in the literature (subsection “Distinct K27M distributions in H3.1 and H3.3 mutant patient-derived cell lines”). In contrast, our model does explain this observation, and we clarify in the Discussion section that in our model H3.3K27M enriched at active sites does not affect EZH2 because the oncohistone only acts locally, and active sites are not sites of PRC2 recruitment (subsection “H3K27 trimethylation is globally reduced in H3.1K27M-bearing cell lines”).

2) Is EZH2 automethylation inhibited by H3K27M?

Yes, we cite in subsection “Distinct K27M distributions in H3.1 and H3.3 mutant patient-derived cell lines” two previous published works that have demonstrated this (PMID: 31488577, 2019; PMID: 31488576, 2019).

Reviewer #2:In this manuscript the authors perform experiments in *Drosophila* and human cell lines to investigate they hypothesis that H3K27M mutation poison PRC2-mediated H3K27 tri-methylation. They provide evidence that, in *Drosophila*, H3.2 inhibition of K27 methylation requires incorporation of H3.2 into nucleosomes and in a separate series of experiments they show that over-expressed H3.3 inhibition of K27 methylation requires proliferation. They then perform Cut&RUN experiments for H3K27M, H3K4me2 and H3K27me3 in several human glioblastoma cell lines. These experiments show different patterns of K27me3 inhibition in H3.3K27M vs. H3.1 K27M harboring cell lines. The paper is generally well-written and easy to follow and the experiments themselves show high quality data. My biggest problem with the manuscript is the incompleteness and incongruency of some of the approaches being cobbled together into universal conclusions that are not entirely warranted. The human results seem much more consistent with replication-dependent vs. independent effects so it is possible that there are artifacts with the fly data (see below) or that flies and humans are not the same.

With respect to similarities between flies and humans, not only are the mechanisms of replication-coupled and replication-independent deposition conserved in detail, but also H3 (H3.2) and H3.3 are 100% identical between the two species. Indeed, the histone chaperones and oncohistone inhibitory effects are also conserved (PMID:25170156). We now cite human data showing that the cell cycle dependence of oncohistone inhibitory effects are also conserved. The importance of our study is that we offer a single explanation for this dependence. The reviewer’s technical concerns are addressed below.

I am also not clear on what are the truly novel findings here. The proliferation-dependence of H3.3 mutant incorporation is intriguing but the one experiment is rather preliminary.

Two mammalian studies (PMID: 23603901, 2013; PMID: 31588023, 2019) have previously shown that progression through at least one cell cycle is necessary for H3.1K27M and H3.3K27M to reduce H3K27me3 levels, although they offer no explanation for this requirement. We now cite these previous results in connection to our results showing a cell cycle requirement in *Drosophila*. However, due to the limitations of mammalian systems, these previous studies were unable to dissect the relationship between cell cycle progression and inhibition of EZH2. We suggest in the Discussion that specifically S-phase progression is necessary for DNAreplication-associated oncohistone deposition and inhibition of EZH2 in Polycomb domains.

It has been known that the H3.1 and 3.3 -associated gliomas have different transcriptomes and epigenomes and that is not really a surprise.

We agree that multiple descriptive studies have shown H3.1 and H3.3-mutant DMGs are associated with different clinical and molecular characteristics. However, an explanation for this difference is completely lacking and remains a critical question in the field. Our work here provides an explanation for these differences. We now emphasize this point in the Results section and Discussion section.

It has been shown that the K27M epitope inhibits EZH2 in vitro, in cell culture, and in H3.3K27M-bearing cell lines.

The first reviewer appreciated that our results showing that inhibition does not occur in solution in vivo is an important finding that extends the previous results referred to here.

Essential revisions:1) The suggestion that H3.3 inhibition of K27me3 requires proliferation is intriguing since the majority of H3.3. is incorporated into histones in a replication-independent manner but it is too preliminary. The authors need to rule out that the effect is not due to the over-expression of H3.3 is driving its incorporation via the CAF1 pathway in their experiments and they need to demonstrate that it is replication directly that is the dependency and not some indirect effect of p21 expression which surely changes many things.

By “indirect” the reviewer might be referring to downstream effects of p21 inhibition of cyclin E. However, the effects of p21 expression in the *Drosophila* eye have been extremely well-studied (PMID: 7481802), and this previous work has demonstrated that p21 expression reduces the number of cells in the eye but does not affect differentiation of those cells. We now show images of these eyes in Figure 2C-F, and the size of these eyes is as expected, with no effect on differentiation of adult ommatidia.

The authors rely strongly on this result in their model in the Discussion, which they present as conserved fly to human, but this experiment in flies is not convincing and whether the same is true in human is not addressed.

The relevant experiments in human cell lines have been published, and demonstrate that cell cycle dependence of oncohistone effect is conserved (PMID: 23603901, 2013; PMID: 31588023, 2019): induced H3.1K27M or H3.3K27M expression in human oligodendrocyte precursor cells or in 293T cells only inhibit EZH2 after progression through at least one cell cycle, although no dissection of cell cycle phases was performed and no explanation for this dependence was presented in either study. We now cite these studies (subsection “Chromatin-bound K27M histone inhibits H3K27 trimethylation in cycling cells”) as the impetus for our experiments in *Drosophila*, since fly retina provides a unique opportunity for probing cell cycle dependencies not afforded by mammalian systems.

Then in light of the human cell results, where the patterns of change in H3K27me3 are more consistent with replication independent and dependent incorporation of the H3.3 variant in hot areas while the 3.1 variant is widespread; I don't understand why they insist that the fly results be incorporated into the model of what they see in the human cell lines.

We have included a new analysis demonstrating that the H3.3K27M oncohistone is in fact widespread through the genome (Figure 3C,F). This is the central point, because this widespread protein must be incorporated during DNA replication, and this analysis should make clear why we are focused on replication. The importance of this finding combined with the fly results is that it provides a simple explanation for why glioma cells with H3.3K27M are less severe than H3.1K27M ones: The inhibition of EZH2 is proportional to the amount of oncohistone incorporated by replication in silenced domains. Note that there is no alternative explanation in the literature for why H3.3K27M oncohistone cells retain H3K27me3 modification at some domains while H3.1K27M ones lack it. We have also included new data (Figure 2C-F) showing that development of the *Drosophila* eye with late K27M oncohistone expression appears normal, indicating that differentiation in this tissue still occurs normally.

2) The authors look at only one variant in each *Drosophila* approach, but draw universal conclusions.

No, we looked at both variants in *Drosophila* in Figure 1 and demonstrated that when overexpressed they act the same. For this reason, we go on with just the H3.3K27M transgene in the *Drosophila* eye. We now highlight this logic in the text in subsection “Chromatin-bound K27M histone inhibits H3K27 trimethylation in cycling cells”.

3) Subsection “G cells maintain PRC2 targeting” – how are we to judge what number of domains "highlights developmental similarity" – we need statistics and negative and positive controls to conclude this. Otherwise all we know is that some are similar and some are different as I suspect would be true in any comparison. Also, for the phrase "akin to stem cells" do the authors refer to embryonic or neural stem cells. If both, then negative controls are very necessary. The total numbers of domains are very different between these two types of stem cells so I'm not even sure that statistics (not discussed) would be helpful but they at least should be stated.

We now limit our statements to state the lack of background H3K27me3 in oncohistone gliomas (subsection “Distinct K27M distributions in H3.1 and H3.3 mutant patient-derived cell lines”) and have removed the comment of developmental similarity to stem cells.

Reviewer #3:In this manuscript, the authors investigate how oncohistones carrying K27M mutations inhibit K27me3, focusing on the role of histone variant subtypes. This work is of high quality and the findings are interesting to the chromatin field. Several of the results in this manuscript cooperates previous work on oncohistones, and the novelty lies in the finding that H3K27M oncohistones only reduce K27me3 level in proliferating cells and the quantitative comparison of H3.1K27M and H3.3K27M cells. The proposed model is interesting, but needs further support. Further dissection of features that distinguish sites that maintain/loose K27me3 in H3.3K27M cells would also strengthen the manuscript. With these revisions the work could be recommended for publication in eLife.Essential revisions:- The authors propose a model that entails replication-coupled incorporation of H3.3 at a low level. However, this work does not provide evidence to support this claim (it does not investigate this point directly) and evidence for this in the literature is weak – Drane et al. show that CAF-1 binds H3.3 in DAXX KO cells and Ray-Gallet et al., explores HIRA dependent GAP filling following replication when CAF-1 function is impaired.

We now provide this evidence in Figure 3C, where we document genome-wide distribution of the K27M epitope in H3.3K27M-bearing cells. These data directly address this concern. In fact, published images of DNA replication and H3.3 in human cells show colocalization of H3.3 at newly replicated DNA, specifically in early S phase but not late-S phase. This was originally shown by Ray-Gallet et al., (PMID: 22195966, 2011) in wild-type cells, and the same group has extended these results with super-resolution microscopy (PMID: 30093638, 2018; Figure 3). These lines of evidence in support of our model are not based on impaired CAF-1 function, unlike the Drane et al., study, which we no longer cite as evidence.

The experiment in Figure 2 shows convincingly that replication is required for H3.3K27M to exert its negative effect on PRC2 and K27me3 levels. However, this could be explained without invoking replication-coupled deposition of H3.3K27M:- Existing K27me3 levels generally remain stable in absence of DNA replication despite reduced PRC2 function (Jadhav et al., 2020). The major requirement for PRC2 function arises with incorporation of large numbers of unmodified histones, likely explaining why H3.3K27M interferes with K27me3 maintenance only in dividing cells.

We agree with the reviewer, and now cite the Jadhav et al. reference where we raise these issues in the Discussion section. The new histones deposited after DNA replication must be modified to maintain epigenomic patterns, and this may explain why blocking S phase also blocks the loss of H3K27me3 in the eye disc. However, a crucial component of our model is that K27M oncohistones must be distributed within Polycomb domains to inhibit the histone methyltransferase, and the mammalian data we have added argues that the toxic fraction of the oncohistone gets there by replication-coupled deposition. We have reworded this section to make clear that we are only showing a requirement for cycling, not replication, and that our findings support our model of replication-coupled deposition.

Moreover, H3.3 incorporation is also expected to increase after replication – via a replication-independent mechanism but due to replication-dependent doubling of the genome.

Analysis of the images shows similar levels of H3.3K27M incorporation in proportion to DNA amount with or without cyclin E inhibition. In contrast we show a quantitatively dramatic drop in H3K27me3 relative to DAPI that is restored when Cyclin E is inhibited (Figure 2I-J).

The authors should either provide evidence to support H3.3 incorporation genome-wide through a replication-coupled mechanism,

We have added data in mammalian cells showing incorporation of H3.3K27M throughout the genome (Figure 3C and F). In previous studies in *Drosophila* cells we showed deposition of H3.3 in S phase cells that coincides with PCNA and with newly incorporated nucleotides (PMID: 12086617, 2002).

or consider to revise the model. They should also investigate the possibility that replication-dependent incorporation of new histones is required to see any effect of PRC2 inhibition in their setup – e.g. would they see loss of H3K27me3 in the *Drosophila* model system in Figure 2 if EZH2 was inhibited in the presence of p21 expression?

This is an interesting experiment, and we were happy to try it. We used an inducible RNAi transgene to knock down E(z) in the *Drosophila* eye just before the last division.

Unfortunately, this only eliminates mRNA and does not affect perduring E(z) protein. There is no effect on H3K27me3 levels in the eye disc with this late knockdown of E(z). We have not included this negative experiment in the manuscript. In contrast, the K27M oncohistone interferes with the activity of existing protein, and so reveals the need for E(z) activity at this stage of development that other strategies cannot.

- The authors show that K27me3 patterns in H3.3K27M cells is more similar to H1 ESCs than VUMC glioma cells. Is this reflecting that the cells are blocked at an earlier developmental stage and if so do gene expression analysis support this notion? what would be the alternative explanation for a specific lack of K27me3 in cluster I and II.

We have removed the comment that DMG epigenomes are similar to that of H1 ESCs, based on criticism of this point from other reviewers.

Is there a difference in H3.3K27M occupancy (histone exchange) across cluster I-IV that may explain why some clusters are more prone to K27me3 loss?

We have added clustered heatmaps showing K27M oncohistone occupancy (Figure 3—figure supplement 2C). There is a low level of H3.3K27M distributed across all clusters, but this does not explain why some clusters have lost H3K27me3 while others have not.

[Editors’ note: what follows is the authors’ response to the second round of review.]

The manuscript has been improved, but there are some remaining issues that need to be addressed before acceptance, as outlined below:In this work, the authors use both *Drosophila* cells/tissues and human patient-derived DMG cells to study two oncohistones H3.1K27M and H3.3K27M. The main conclusions are: (1) Inhibition of oncohistone on K27me3 only happens in cis when mutant histones are incorporated into chromatin and only in dividing cells. (2) They showed that the distinct deposition modes of H3.1 and H3.3 determine their genome distribution with H3.1K27M showing a more global distribution while H3.3K27M having more localized enrichment. (3) Neither H3.1K27M nor H3.3K27M oncohistone interferes with PRC2 binding.The experimental strategies combined individual cell assay such as immunostaining and genomic analysis such as CUT&RUN, the later gave out clean results with high S/N ratio. The results are presented in a logic manner and the main conclusions are made carefully (except the third one, see Essential revision #4 below). However, for several places, the presentation of the results needs revision or improvement. Overall, this work should be appropriate for eLife after revisions.Essential revisions:1) Figure 1: In transfected *Drosophila* S2, if cells are not allowed to progress "2-3 cell cycles", do they lose or maintain H3K27me3? This would be good to include as a control, and for the conclusion cell cycle progression is required.

This experiment requires control of cell cycle progression and of histone construct induction. This is difficult to do in cell culture, which is why we turned to the eye system. Nevertheless, we used a set of heat-shock-inducible constructs (similar to the set of constitutive constructs we described in the manuscript) to lipofect S2 cells, expand those cells, then arrest cells in G1 phase of the cell cycle with the small molecule inhibitor mimosine, and then induce expression of histone constructs. With H3.3K27M, cell cycle inhibition by mimosine switches the histone deposition to an entirely replication-independent pattern (the puncta represent active rDNA genes (PMID: 12086617), and reduces inhibition of H3K27 trimethylation [Author response image 1]). With H3K27M, cell cycle inhibition similarly reduces heat-shock-induced H3K27 trimethylation. These results visually resemble what we have seen in eye discs, but quantification shows that induced inhibition in both cycling and arrested cells is marginal, and for this reason we have not included S2 cell cycle arrest in the manuscript.

**Author response image 1. respfig1:** H3K27M fails to inhibit H3K27 trimethylation in G1-arrested cells. S2 cells were lipofected with heat-shock-inducible epitope-tagged histone (red) constructs and allowed to progress through cell cycles or arrested with mimosine for 24 hrs before construct induction or 1 hr at 37°, and then allowed to recover for 8 hrs at 25°. Cells were fixed and stained with anti-H3K27me3 antibody (green) and DAPI (blue). Nuclei with epitope-tagged histone are marked with a yellow asterisks. H3K27M and H3.3K27M oncohistones inhibit H3K27 trimethylation in cycling cells, but are less inhibitory in mimosine-arrested cells.

2) Figure 2 uses eye imaginal discs to study the effect of oncohistones. This is a great design as cells either exit cell cycle or undergo active mitosis can be visualized all together. A few suggestions on this figure: (1) Only H3.3K27M is studied here, why leaving H3K27M out for this experiment?

Our cell line work (Figure 1 and Author response image 1) showed that overexpression of H3K27M and H3.3K27M is equally effective at inhibiting H3K27 trimethylation.

(2) Figure 2A-B need to improve image resolution, it is quite blurry. (3) The H3S10 phosphorylation immunostaining signal is unclear, it is hard to tell the synchronously dividing cells right posterior to the MF. It would be better to show a better resolution image with this channel separate in gray scale.

We have replaced Figure 2A,B with better-quality images with the requested separate channel.

(4) In Figure 2 legend, it is said that "at least 10 eye discs were examined for each genotype". In the figure, only one eye or eye disc was shown. First, N=10 is not a large sample size; second, quantification on them should be shown or mentioned in the text, such as what percentage of the samples show what phenotype, etc.

We added quantification of 10 eye discs for each genotype by measuring signal intensities in the anterior and posterior portions of each disc, and added this quantification to Figure 2M.

3) Figure 3E: Further analysis on H3.3K27M-enriched promoters should be informative to understand the molecular mechanisms of this oncohistone.

Thank you for this recommendation, we now include gene lists for loci that are enriched for H3.3K27M signal. As expected, we find enrichment of H3.3K27M at transcribed genes, including *SOX2*, MYC, OLIG2 and MYC, and we discuss these results in subsection “Distinct K27M distributions in H3.1 and H3.3 mutant patient-derived cell lines”.

4) For this conclusion- "Neither oncohistone interferes with PRC2 binding." The authors examined binding of SUZ12 and MTF2 in Figure 5, however, the key PRC2 component that should be studied is EZH2, the methyl-transferase. It would be very informative to examine EZH2's binding here.

We appreciate this point and have attempted to profile EZH2 using multiple different antibodies. However, while EZH2 is enriched at H3K27me3 peaks, the signal-to-noise ratio is low, as demonstrated in the heatmaps below where EZH2 CUT&RUN signal was mapped on H3K27me3 domains. We only see EZH2 signal over the strongest H3K27me3 domains (Cluster 3). The noise in this profiling is due to the anti-EZH2 antibody, as we observe similar high background in the H3WT cell lines. However, EZH2 has been previously profiled in H3.3K27M-mutant cells by Piunti et al., (2017), where it coincides with SUZ12 and H3K27me3 and is anti-correlated with K27M.

**Author response image 2. respfig2:** Enrichment of EZH2 in H3K27me3 domains. CUT&RUN signal for EZH2 (blue) in differential H3K27me3 domains defined in Figure 4A.